# Meta-Forecasting by combining Global Deep Representations with Local Adaptation

## Abstract

While classical time series forecasting considers individual time series in isolation, recent advances based on deep learning showed that jointly learning from a large pool of related time series can boost the forecasting accuracy. However, the accuracy of these methods suffers greatly when modeling out-of-sample time series, significantly limiting their applicability compared to classical forecasting methods. To bridge this gap, we adopt a meta-learning view of the time series forecasting problem. We introduce a novel forecasting method, called Meta Global-Local Auto-Regression (Meta-GLAR), that adapts to each time series by learning in closed-form the mapping from the representations produced by a recurrent neural network (RNN) to one-step-ahead forecasts. Crucially, the parameters of the RNN are learned across multiple time series by backpropagating through the closed-form adaptation mechanism. In our extensive empirical evaluation we show that our method is competitive with the state-of-the-art in out-of-sample forecasting accuracy reported in earlier work.

## 1 Introduction

Time series (TS) forecasting is of fundamental importance for various applications like marketing, customer/inventory management, and finance (Petropoulos et al., 2020). Classical examples are forecasting the number of daily sales of a product over the next few weeks or the energy production needs in the next few hours. Accurate TS forecasting results in better down-stream decision making with potentially large monetary implications (e.g., Seeger et al. (2016); Faloutsos et al. (2019)).

From a machine learning perspective, each TS represents a forecasting task. Conventional approaches for TS forecasting have typically been local, i.e. each TS/task is modeled independently by a forecasting model with relatively few parameters (see Hyndman and Athanasopoulos (2018) for an introductory overview). Despite the modest amount of data used to train local TS forecasting models, they are effective in practice. They have only recently been outperformed by global deep learning strategies, which jointly train a deep neural network on a large set of related TS/tasks. Global models are designed to work well on the set of TS they are trained on, but they perform poorly on out-of-sample TS, i.e. TS which are not present in the training set. For example, Oreshkin et al. (2020) show that DeepAR (Salinas et al., 2020) trained on the M4 dataset (Makridakis et al., 2020) performs poorly on the M3 dataset. Global-local approaches (Sen et al., 2019; Smyl, 2020; Wang et al., 2019), such as the M4 competition winner (Smyl, 2020), exhibit a greater level of specialization as they learn parameters that are shared by all TS in the training set, as well as parameters specific to each TS. However, global-local models are still not able to handle out-of-sample TS as both types of parameters are learned jointly on a large training set of related TS in a multi-task fashion.

In this work, we tackle the problem of out-of-sample TS forecasting by transferring knowledge from a set of TS, called the source dataset, to another set of TS, called the target dataset. We assume that source and target datasets share some underlying structure which makes transfer learning possible, although they may contain TS from different domains. Our work can be seen as an instance of *meta-learning* (Schmidhuber, 1987; Ravi and Larochelle, 2016; Finn et al., 2017), whose goal is to leverage a pool of related tasks to learn how to adapt to a new one with little data. We will refer to a forecasting method performing well in this scenario as a meta-forecasting method.[1] Such models

---

[1]This was called zero-shot transfer learning by Oreshkin et al. (2020).

could in principle be trained on a large set of TS and still produce accurate and fast predictions when applied to out-of-sample TS, potentially combining the inference speed and accuracy of deep learning models with the ease-of-use of classical local models.

Our meta-forecasting method produces one-step ahead forecasts by combining learned representations with a differentiable closed-form adaptation mechanism inspired by the few-shot image classification method proposed by Bertinetto et al. (2018). Specifically, we propose a class of models which we call Meta Global-Local Auto-Regressive models (Meta-GLAR). Meta-GLAR models compute point forecasts for a single TS in three steps. First, the TS is passed through a representation model to obtain a representation for each time step. Second, a local (i.e. TS-specific) linear model is learned in closed-form by solving a ridge regression problem mapping the representations to the observed fraction of the TS. Lastly, the local linear model is applied to the global representation to compute the final predictions. Crucially, forecasts are computed in the same way also during training, where we backpropagate through the closed-form adaptation step to learn the representation parameters globally, i.e. across multiple TS. Hence, we can learn a representation which works well in combination with an efficient closed-form local adaptation.

We use the RNN backbone in DeepAR (Salinas et al., 2020) for the representation. However, one can transform any neural forecasting method in a meta-GLAR one by substituting the last global linear layer with the closed-form adaptation during both training and prediction. We also stress that, since our method adapts locally also during training, it is fundamentally different from fine-tuning the last layer on each TS, which we show to perform significantly worse. This is in contrast to recent results for few-shot image classification where fine-tuning the last layer is demonstrated to outperform many modern meta-learning methods (Tian et al., 2020).

Our main contributions can be summarized as follows.

- We propose a novel meta-learning method for TS forecasting that is suitable for out-of-sample TS forecasting. Meta-GLAR significantly improves the accuracy on out-of-sample TS forecasting (i.e., transfer setting) over a global neural forecasting models such as DeepAR, which employs the same RNN backbone.

- Our meta-forecasting method is competitive with classical local methods, for example beating the winner of the M3 competition, as well as NBEATS (Oreshkin et al., 2020), the state-of-the-art method for out-of-sample TS forecasting, while having substantially fewer parameters than the latter method.

- We perform an extensive ablation study which shows that the closed-form adaptation, the RNN backbone, and the use of iterated forecasts during training are needed to achieve the best performance. Furthermore we show that Meta-GLAR enjoys similar time and memory costs compared to a global one-step ahead RNN with the same backbone architecture, while converging faster and achieving better accuracy.

## 2 RELATED WORK

Meta-learning has received considerable attention in recent years and several models have been developed primarily for few-shot image classification. Notable examples are Ravi and Larochelle (2016); Finn et al. (2017); Snell et al. (2017); Nichol et al. (2018); Sung et al. (2018); Bertinetto et al. (2018). These methods work by adapting to the task at hand before making predictions. Differently from fine-tuning, this adaptation is performed also during a (meta) training procedure over a large set of tasks to learn the (meta) parameters of the model. However, realistic datasets with the high number of tasks needed by meta-learning methods are rare, hence in commonly used benchmarks like mini-imagenet (Vinyals et al., 2016), each classification task is constructed by randomly selecting a small set of classes and related images from a large single-task dataset. This construction is artificial and departs from real-world scenarios. Recently, the work by Tian et al. (2020) showed that training a neural network on the original single-task dataset and fine-tuning only the last layer on new tasks outperforms many modern meta-learning methods for few-shot image classification. By contrast, popular TS forecasting datasets like M4 fit more naturally into the meta-learning framework, since they already contain a large number of TS/tasks (up to $10^5$ for M4).

Our method relies on a differentiable closed-form solver to perform the local (or TS-specific) adaptation. Meta-learning is achieved by solving a task-specific ridge regression problem that maps a

deep representation to the target TS in closed-form, while the parameters of the representation are learned by backpropagation through the solver. Aside from the original application in few-shot image classification (Bertinetto et al., 2018), differentiable closed-form solvers have been used for other few-shot problems like visual tracking (Zheng et al., 2019), video object segmentation (Liu et al., 2020), spoken intent recognition (Mittal et al., 2020) and spatial regression (Iwata and Tanaka, 2020), while we are not aware of any application in forecasting.

Meta-learning in the context of TS forecasting has originally been synonym with model selection or combination of experts (see e.g. Collopy and Armstrong (1992); Lemke and Gabrys (2010); Talagala et al. (2018)). This class of methods builds a meta-model which uses TS features to select the best performing model or the best combination of models to apply to a target TS. One drawback of these methods is that the features are usually manually designed from the data at hand and that the same set of features does not transfer well to other applications. Laptev et al. (2018) train an LSTM neural network Hochreiter and Schmidhuber (1997) on the source TS dataset and then its last layers are fine-tuned separately on each target TS. This approach overcomes the problems related to human designed features since the input of the network are just the previous TS observations. However, retraining the last layers of the network for each TS can be expensive, especially when dealing with a large number of TS. Additionally, their fine-tuning procedure requires the selection of hyperparameters like learning rate and number of steps of the optimizer. By contrast, our approach adapts only the last linear layer in closed-form, requiring a small increase in compute and no additional hyperparameters compared to a standard neural forecasting model, while outperforming the simple fine tuning approach.

More recently, the NBEATS model has shown strong performance both in the standard (Oreshkin et al., 2019) and in the meta-learning (Oreshkin et al., 2020) setting. This multi-step ahead method uses a residual architecture (He et al., 2016) which takes past observations of a single TS as input and outputs point predictions over the whole forecast horizon. Thanks to the residual connections, a forward pass of the network allows it to implicitly adapt to the input out-of-sample TS. However, the final performance is achieved using a large ensemble and the number of parameters of the model is quite large even when the residual blocks share the same parameters. Our method, although also using ensembles, achieves good accuracy on out-of-sample TS forecasting with significantly less parameters.

Finally, Iwata and Kumagai (2020) consider the few-shot TS forecasting setting where each task is formed by a small group of closely related TS. Their method, which combines LSTMs and attention, uses the TS in the support set of the task to compute the one-step ahead forecasts for each TS in the query set. This is different from our approach, where we do not exploit the other TS in the target dataset to compute predictions and we view each TS as a separate task. Our method can be extended to the case considered by Iwata and Kumagai (2020) by performing the closed-form adaptation on all the TS in the task instead of just on one. We leave this for future work.

## 3 PROBLEM FORMULATION

We consider the setting studied by Oreshkin et al. (2020), where a forecasting method can learn global parameters on a source TS dataset $\mathcal{D}_\mathcal{S}$ to produce accurate forecasts for an out-of-sample TS which belongs to another, target TS dataset $\mathcal{D}_\mathcal{T}$. The model can only adapt locally to each TS in $\mathcal{D}_\mathcal{T}$, i.e. it cannot use information from the other TS in $\mathcal{D}_\mathcal{T}$. We view each TS as a task. Hence, our setting fits a meta-learning framework where $\mathcal{D}_\mathcal{S}$ is the meta-training set and $\mathcal{D}_\mathcal{T}$ is the meta-testing set.

We will denote a single TS, as a tuple $(\mathbf{z}, \mathbf{x})$ where $\mathbf{z} = [z_1, \ldots, z_T] \in \mathbb{R}^T$ are the (target) observations and $\mathbf{x} = [\mathbf{x}_1, \ldots, \mathbf{x}_T] \in \mathbb{R}^{T \times p}$ is the matrix of covariates. We will denote with $t_0 \in \mathbb{N}$ the split point which divides the context window (or past) $\mathbf{z}_{1:t_0} = [z_1, \ldots, z_{t_0}]$, $\mathbf{x}_{1:t_0} = [\mathbf{x}_1, \ldots, \mathbf{x}_{t_0}]$ from the forecast horizon (or future) $\mathbf{z}_{t_0+1:T} = [z_{t_0+1}, \ldots, z_T]$, $\mathbf{x}_{t_0+1:T} = [\mathbf{x}_{t_0+1}, \ldots, \mathbf{x}_T]$. We also denote with $H = T - t_0$ the length of the forecast horizon. We view each TS as a supervised learning task with training set $\{(\mathbf{x}_t, z_t)\}_{t=1}^{t_0}$ and test set $\{(\mathbf{x}_t, z_t)\}_{t=t_0+1}^{T}$ where an example is the covariates-target pair $(\mathbf{x}_t, z_t)$. We assume that the covariates vector $\mathbf{x}_t$ can contain some of the previous observations $z_{t-1}, z_{t-2}, \ldots$ as time-lagged values (or time-lags). Differently from standard supervised learning tasks, we cannot assume that the examples are independent due to the temporal dependency.

The goal of TS forecasting is to compute predictions $\hat{\mathbf{z}}_{t_0+1:T}$ for the observations in the forecast horizon $\mathbf{z}_{t_0+1:T}$ using the covariates $\mathbf{x}$ and the observations in the context window $\mathbf{z}_{1:t_0}$. In this work

we focus on the accuracy of the predictions which can be measured for example with the sMAPE metric (Hyndman and Koehler, 2006):

$$\text{sMAPE} = \frac{1}{|\mathcal{D}|} \frac{200}{H} \sum_{i=1}^{H} \frac{|z_{t_0+i} - \hat{z}_{t_0+i}|}{|z_{t_0+i}| + |\hat{z}_{t_0+i}|}. \tag{1}$$

## 4 META GLOBAL-LOCAL AUTO-REGRESSIVE FORECASTING

The forecasting model we prosose is non-linear, auto-regressive and computes one-step ahead forecasts, i.e. for each time step $t$ in the forecast horizon, the model outputs a point forecast $\hat{z}_t$ as a non-linear function of the covariates $\mathbf{x}_{1:t}$ and the observations in the context $\mathbf{z}_{1:t_0}$. The model generates forecasts over the forecast horizon using iterated (or recursive) forecasts (Salinas et al., 2020). Hence, the model will use new covariates vectors which contain previous forecasts in place of the missing time-lags for the time-steps in the horizon. These new vectors are constructed together with the forecasts in an iterative fashion from the first to the last point in the horizon. By contrast, multi-step ahead approaches like NBEATS compute forecasts over the whole horizon directly.

Starting from $\mathbf{h}_0 = 0 \in \mathbb{R}^d$, the Meta Global-Local Auto-Regressive (Meta-GLAR) forecasting model computes point predictions as follows:

$$\mathbf{h}_t := h(\mathbf{x}_t, \mathbf{h}_{t-1}; \mathbf{\Theta}), \quad \forall t \in [1:T], \tag{2}$$

$$\hat{z}_t := \mathbf{w}_{\text{opt}}^\top \mathbf{h}_t, \quad \forall t \in [t_0+1:T], \tag{3}$$

where $\mathbf{h}_t$ and $\hat{z}_t$ are respectively the $d$-dimensional representation and point forecast at time $t$. Here, $h$ is a recurrent function providing an appropriate representation for TS. While in this work we choose to use the LSTM architecture used by DeepAR (Salinas et al., 2020), our approach is not limited to it and could use other representations such as the one in Franceschi et al. (2019). $\mathbf{\Theta}$ and $\mathbf{w}_{\text{opt}}$ are respectively the global and local parameters.

**Estimating the local parameters.** The local parameters $\mathbf{w}_{\text{opt}}$ are learned explicitly and in closed-form on the training set of a single TS/task, which contains the data in the context window. This is done by solving a ridge regression problem having as inputs and targets respectively the observations $\mathbf{z}_{1:t_0}$ and the representation vectors $\mathbf{h}_{1:t_0}$:

$$\mathbf{w}_{\text{opt}} := \arg\min_{\mathbf{w} \in \mathbb{R}^d} \sum_{t=1}^{t_0} (\mathbf{w}^\top \mathbf{h}_t - z_t)^2 + \gamma \|\mathbf{w}\|^2 \tag{4}$$

$$= (\mathbf{h}_{1:t_0}^\top \mathbf{h}_{1:t_0} + \gamma I)^{-1} \mathbf{h}_{1:t_0}^\top \mathbf{z}_{1:t_0}, \tag{5}$$

where $\gamma \in \mathbb{R}^+$ is the regularization parameter. Aside from the training procedure, the key difference with a global neural forecasting model is that $\mathbf{w}_{\text{opt}}$ is a local parameter. Albeit in closed-form, computing (5) requires either a $d \times d$ matrix inversion or solving a linear system and has a cost of $\Omega(d^2 t_0)$. Note, however, that $d$ is usually small (e.g. 20-50) and the cost of computing the representation vectors $\mathbf{h}_{1:t_0}$ is usually far higher, especially for large $t_0$ and when $h$ is an RNN.

**Estimating the global parameters.** Similarly to other neural forecasting models, the global parameters $(\mathbf{\Theta}, \gamma)$ are trained instead by solving the following minimization problem on the source dataset $\mathcal{D}_{\mathcal{S}}$ using a stochastic gradient method:

$$\min_{\mathbf{\Theta}, \gamma} \sum_{(\mathbf{z}, \mathbf{x}) \in \mathcal{D}_S^{\text{tr}}} \mathcal{L}(\hat{\mathbf{z}}_{t_0+1:T}, \mathbf{z}_{t_0+1:T}), \tag{6}$$

where $\mathcal{L}$ is a regression loss suitable for forecasting (typical choices are (scaled) MAE or sMAPE) and $\mathcal{D}_S^{\text{tr}}$ is the training split of $\mathcal{D}_{\mathcal{S}}$. We note that the loss is computed over the test set of each TS/task in $\mathcal{D}_S^{\text{tr}}$, which contains the data in the forecast horizon, not used to compute $\mathbf{w}_{\text{opt}}$. This prevents the overfitting of $\mathbf{w}_{\text{opt}}$ and allows to learn global parameters which generalize well to each TS/task. We use iterated forecasts also during training even if the true target observations in the forecast horizon are available in this phase and show that this increases the performance, in line with Bengio et al. (2015), even without scheduled sampling. $\mathcal{D}_S^{\text{tr}}$ is typically constructed by removing the last H observations and covariates from each TS in $\mathcal{D}_{\mathcal{S}}$. The last H observations are used to test the in-sample performance of the method on $\mathcal{D}_{\mathcal{S}}$.

**Gradient Contributions.** Recall that the predictions ($t > t_0$) are given by $\hat{z}_t = \mathbf{w}_{opt}^\top \mathbf{h}_t$, where $\mathbf{w}_{opt}$ is the solution to the least squares problem for the past time points $t \leq t_0$ and thus depends on the past representations $\mathbf{h}_{1:t_0}$. Then, the gradient of the loss with respect to the global parameters $\mathbf{\Theta}$ has two components:

$$\frac{\mathrm{d}\mathcal{L}}{\mathrm{d}\mathbf{\Theta}} = \sum_{t=t_0+1}^{T} \frac{\mathrm{d}\mathcal{L}}{\mathrm{d}\hat{z}_t} \left[ \frac{\mathrm{d}\mathbf{h}_t}{\mathrm{d}\mathbf{\Theta}} \mathbf{w}_{\mathrm{opt}} + \sum_{\tau=1}^{t_0} \frac{\mathrm{d}\mathbf{h}_\tau}{\mathrm{d}\mathbf{\Theta}} \frac{\mathrm{d}\mathbf{w}_{\mathrm{opt}}}{\mathrm{d}\mathbf{h}_\tau} \mathbf{h}_t \right]. \tag{7}$$

Hence, the first component is the direct gradient of the loss with respect to the representations in the prediction range $\mathbf{h}_{t_0+1:T}$. The second component is an indirect contribution from the representations in the context $\mathbf{h}_{1:t_0}$, which influence $\mathbf{w}_{opt}$.

Since $\mathbf{w}_{\mathrm{opt}}$ is obtained in closed-form and differentiable, we can compute the gradient of the loss using automatic differentiation. Replacing the Mean Squared Error in (4) with another loss, such as the one used in (6), usually leads to $\mathbf{w}_{\mathrm{opt}}$ not being in closed-form. However, if (4) is still a convex minimization problem or conic program, our approach still applies (Agrawal et al., 2019), e.g., by using iterative reweighted least squares. If instead (4) becomes non-convex, one could still backpropagate through a few gradient descent updates as in Raghu et al. (2019).

Intuitively, the RNN is trained to generate time dependent (and TS specific) features $\mathbf{h}_t$ that work well for the local linear forecasting model. In principle, this design allows the model both to learn the common traits of the TS in $\mathcal{D}_\mathcal{S}$ and to adapt to individual TS, combining the properties and benefits of both global and local models. Additionally, since $\mathbf{w}_{\mathrm{opt}}$ is learned explicitly on each TS (also in $\mathcal{D}_\mathcal{T}$) before computing predictions, the model should adapt better to potential dataset shifts between $\mathcal{D}_\mathcal{S}$ and $\mathcal{D}_\mathcal{T}$. This is in contrast to other global-local models like the M4 competition winner (Smyl, 2020), which learn the local parameters together with global ones only during training.

## 5 EXPERIMENTS

In all our experiments, we train on a *source* dataset and measure predictive performance on a *target* dataset. We stress that when source and target dataset are not the same, the model computes the forecasts for each TS in the target dataset without access to the other TS in the target dataset. This is different from domain adaptation settings where source and target dataset are available at training time. To compare with NBEATS, we use the same source-target combinations reported by Oreshkin et al. (2020): we train on one of the M4 datasets and use datasets with same frequency as target datasets whenever possible (see Table 6 in the Appendix). M4 contains a large number of diverse TS, thus there is higher chance for it to share some properties with other datasets. An ideal source dataset would contain a very diverse set of time series from many real-world applications from different domains (and different frequencies).

**Transfer from M4.** In Table 1, for both Meta-GlAR and DeepAR and for each frequency of the M4 dataset we repeat the following model selection. We evaluate 200 random hyperparameter configurations and pick the 10 configurations with the lowest sMAPE on a subset of $8K$ training TS. The hyperparameters for the random search are described in the Appendix (Table 8). A single Meta-GLAR model (Meta-GLAR-top10) is competitive with Theta and ARIMA and always outperforms DeepAR. Ensembling can improve performance for both DeepAR and Meta-GLAR. An ensemble of 10 Meta-GLAR models (Meta-GLAR-ens10) outperforms NBEATS, which consists in an ensemble of 90 models, on ELECTR, as well as DeepAR trained on the target dataset on ELECTR and M3. NBEATS is still superior on TRAFF and TOURISM while the gap is much closer on M3. As shown in Table 2, our method has a substantially lower global parameter count than NBEATS: from 35 to 1000 times less parameters for the monthly frequency. We did not notice performance improvements of our method by increasing the number of layers or the hidden dimensions. This is a common behavior of one-step-ahead RNN based forecasting models (see e.g. Salinas et al. (2020)), whose accuracy does not increase much beyond a certain model size. By contrast, NBEATS is a multi-step ahead residual network that computes a sequence of predictions in a single forward pass and hence benefits from a larger number of parameters. More detailed results are in Appendix A.1.

**Ablation analysis.** In Figure 1, we perform an ablation analysis to evaluate the benefits of different elements of our method (Meta+RNN+ITF). These are the closed-form adaptation (Meta), the RNN network for the representation and the use of iterated forecast during training (ITF). We perform

Table 1: Out-of-sample TS forecasting. Global parameters of all models are trained on M4 except for the entries with *target* in the name, which are instead trained on the target dataset. We evaluate the 10 selected models both individually (top10), reporting mean $\pm$ 95% confidence interval of the performance metric, and as an ensemble which takes the median of the predictions (ens10). Best performance for each column is in **bold** and does not take into account the models with target in the name since they use all the TS in the target to learn the global parameters.
* as reported in Salinas et al. (2020) (DeepAR-target) and Oreshkin et al. (2020) (all the others).

| | ELECTR (ND) | TRAFF (ND) | M3 (sMAPE) | TOURISM (MAPE) |
|---|---|---|---|---|
| Theta[*] | 0.080 | 0.170 | 13.015 | 20.878 |
| ARIMA[*] | **0.067** | **0.145** | 14.005 | 20.959 |
| DeepAR-target[*] | 0.070 | 0.170 | 12.671 | 19.276 |
| NBEATS-target[*] | 0.067 | 0.114 | 12.374 | 18.523 |
| DeepAR[*] | 0.150 | 0.360 | 14.767 | 24.787 |
| NBEATS[*] | 0.090 | 0.150 | **12.441** | **18.828** |
| DeepAR-top10 | $0.211 \pm 0.037$ | $0.407 \pm 0.064$ | $13.297 \pm 0.101$ | $23.920 \pm 0.631$ |
| DeepAR-ens10 | 0.199 | 0.321 | 12.884 | 22.889 |
| Meta-GLAR-top10 | $0.079 \pm 0.006$ | $0.245 \pm 0.014$ | $12.746 \pm 0.048$ | $21.254 \pm 0.247$ |
| Meta-GLAR-ens10 | **0.067** | 0.188 | 12.509 | 20.095 |

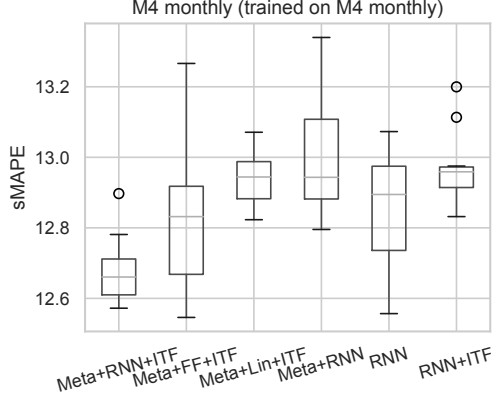 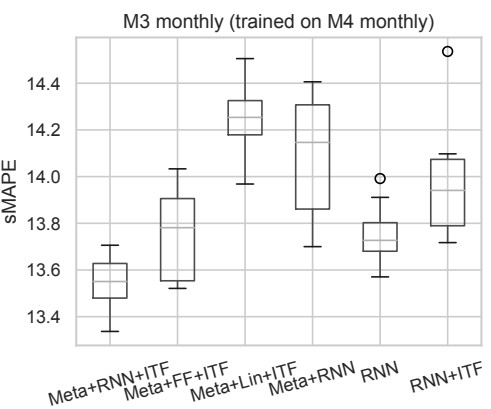

Figure 1: Ablation study for Meta-GLAR (Meta+RNN+ITF). Ablated elements are the representation function, which can be Linear (Lin) a feed-forward (FF) or a recurrent (RNN) neural network, the closed-form adaptation from meta-learning (Meta) and iterated forecasts during training (ITF). Each box plot shows statistics for the 10 over 100 random search runs with lowest sMAPE computed on 8K random training TS.

combinations of the following ablations. Replacing the closed-form adaptation with a global linear layer (no Meta in the name). Using the observations in the horizon to compute predictions during training instead of iterated forecasts (no ITF in the name). Replacing the RNN with a stateless feed-forward model (FF) and a linear (Lin) model. The results show that all the three ingredients are necessary to achieve the best performance. In addition, we note that in this scenario, using iterated forecasts during training without the closed-form adaptation (RNN+ITF) performs worse than the auto-regressive RNN model. Replacing the last global linear layer with the closed-form adaptation gives our model more "memory" even when we use a feed-forward or linear backbone, since the last linear layer is computed using past representations in the context, while an autoregressive stateless model only uses the last representation in the context. Hence, it is interesting to see that the RNN backbone performs the best even in this setting. Ablation on the hourly frequency is in Appendix A.2.

**Comparison with fine-tuning.** Recently, Tian et al. (2020) criticized meta-learning approaches in computer vision and achieved state-of-the-art results for few-shot image classification only by

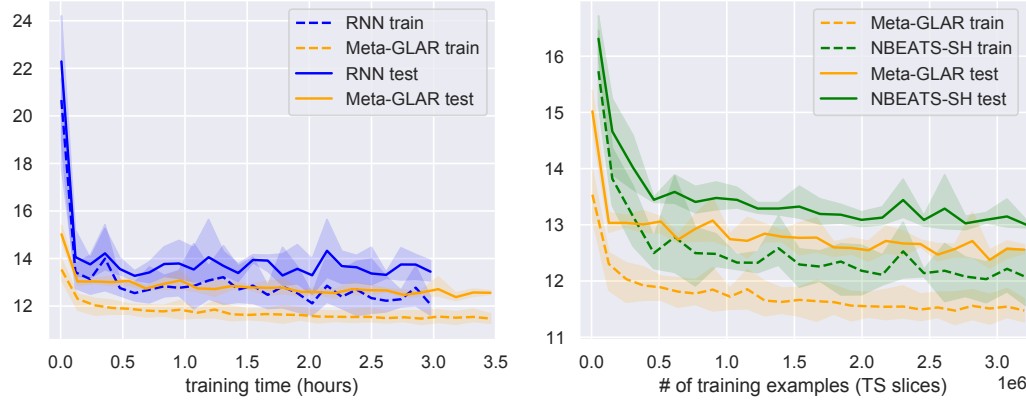

Figure 2: mean and max − min (shaded regions) sMAPE over training computed on 8K random time series from the train (dashed lines) and test (solid lines) sets of the source dataset (M4 monthly). Results are for 4 runs of each method where we only varied the seed controlling the network initialization and the training minibatches. All methods are optimized using the same loss function (sMAPE). Training consists in $25K$ ADAM steps with minibatch size 128 for Meta-GLAR and RNN and 3500 steps with minibatch size 1024 for NBEATS-SH.

Table 2: Number of global parameters of the single models for the monthly frequency. From the GluonTS implementation. NBEATS-SH and NBEATS-NSH contain 30 residual blocks with (SH) and without (NSH) shared weights as described in Oreshkin et al. (2020).

| Meta-GLAR/RNN/DeepAR | NBEATS-SH | NBEATS-NSH |
|:---:|:---:|:---:|
| $\approx 25K$ | $\approx 880K$ | $\approx 25M$ |

learning a good representation and then using it with a local model at prediction time, avoiding any local learning or adaptation during training. We explored this approach by training a standard global RNN model on the source dataset. For prediction on the target dataset, we then kept the network weights fixed, but replaced the last layer with the closed-form adaption layer (with $\gamma = 1$), corresponding to the setting of Tian et al. (2020) of a fixed global representation trained without meta learning. This method reaches a median sMAPE[2] of 14.16, 15.40 (ITF) and 14.13, 15.74 (ITF) for M4 and M3 monthly respectively, which is far higher than all the other methods in Figure 1. While we used the closed-form adaption layer in our experiments, the solution is equivalent to training the last layer from scratch using SGD. We also tried a variant of this experiment where we use the regularization $\gamma\|w - w_{\text{global}}\|$ where $w_{\text{global}}$ are the parameters of the global last layer. This is equivalent to gradient descent fine-tuning of the last layer starting from $w_{\text{global}}$. This variant reaches 14.86 and 16.06 median sMAPE respectively on M4 and M3 monthly (no ITF). These experiments show that in the forecasting setting it is important to meta-learn the representation parameters such that the local model works well, in contrast to the results of Tian et al. (2020).

**Learning curves.** In Figure 2, we compare the performance during training of our method with that of NBEATS with shared weights (NBEATS-SH) and the global RNN method from our ablation analysis. Meta-GLAR and RNN use the same model and training hyperparameters, while NBEATS uses an 8 times larger minibatch size (the same used by Oreshkin et al. (2020)). The gap in performance is large at the beginning of the training. We believe this is due to the local closed-form adaptation, which allows to have greater performance even with random global parameters. We also note that the performance of Meta-GLAR is more stable during training and that the training time (on CPU) is only 17% higher than that of the non-adaptive RNN. NBEATS is significantly faster to train: 1.38 hours for 300 epochs with batch size 1024. This is expectd since NBEATS computes all predictions in a single forward pass and thus benefits more from parallel computation. The prediction time for Meta-GLAR

---

[2]We computed the median on the 10 over 100 random search runs and with lowest sMAPE on a subset of $8K$ training TS.

is only marginally higher than that of a the non-adaptive RNN, i.e. it takes 3:21 minutes instead of 3:19 to forecast the full M4 test set, while NBEATS takes 3:15 minutes (we set the minibatch size to 1K for all methods during prediction). Note also that the non-adaptive RNN training and prediction times similar to DeepAR since both have the same RNN backbone.

## 5.1 EXPERIMENT DETAILS

**Benchmark datasets.** We use five datasets in our experiments[3]. ELECTR contains hourly time series of the electricity consumption of 370 customers (Dua et al., 2017). TRAFF time series measures hourly occupancy rate, between 0 and 1, of 963 San Francisco car lanes (Dua et al., 2017). TOURISM is a collection of yearly, monthly and quarterly series of indicators related to tourism activities (Athanasopoulos et al., 2011). M3 and M4 are two collection of time series that were used in recent forecasting competitions (Makridakis and Hibon, 2000; Makridakis et al., 2020). M3 comprises 645 yearly, 756 quarterly, 1428 monthly as well as 174 TS without time granularity (M3-OTHER). Finally M4 contains 100K time series in total of yearly, quarterly, monthly, weekly, daily and hourly granularities. Both M3 and M4 contain heterogeneous TS types from business, financial and economic domains. All datasets are accessed through the GluonTS library (Alexandrov et al., 2020). Following Oreshkin et al. (2020), to compare with previous result from the literature we report the commonly used metrics for each dataset: sMAPE for M4 and M3; ND, which corresponds to the P50 loss in Salinas et al. (2020), for ELECTR and TRAFF; and MAPE for TOURISM.

**Baselines.** We report the top baselines from Oreshkin et al. (2020), namely Theta (local) (Assimakopoulos and Nikolopoulos, 2000), auto-ARIMA (local), NBEATS and DeepAR (Salinas et al., 2020). We also report the results from DeepAR from Salinas et al. (2020) and train and evaluate the DeepAR model using a procedure similar to our method. We recall that local methods do not require training on the source dataset and their parameters are estimated directly on each TS of the target dataset. Fine-tuning the whole network on each TS and other meta-learning methods used primarily in computer-vision like MAML (Finn et al., 2017), REPTILE (Nichol et al., 2018) or ANIL (Raghu et al., 2019) can in principle be applied to TS forecasting. However, all require an additional hyperparameter: the number of steps of gradient descent used to learn the local parameters, which can't be easily meta-learned. Furthermore, fine-tuning the whole network on each TS, MAML and REPTILE have as local parameters all the network weights, hence they would be significantly slower and occupy more memory than our method during prediction and, for MAML and REPTILE also during training. On the other hand, ANIL adapts only the last layer and thus should have a similar time and memory cost than our method. Applying these methods to time-series forecasting is an interesting future work.

**Models input/output.** Like many RNN forecasing models, we provide previous observations as input to the models (time-lags). The lags for a given time-frequency are detailed in Table 7 of the appendix. In addition, we scale/descale the input time-lags and final output of the model by dividing/multiplying by a scaling coefficient which is the average of the absolute value of the observations in the context window.[4] We also use the log of the scaling coefficient and the "age", i.e. the distance from the first observation in TS as additional covariates. Forecasts below zero are set to zero, since all the datasets contain positive values. The NBEATS model applies a similar scaling but it takes as input to the residual network just the observations in the context window, without using covariates. In our setting it is counterproductive to use the index of the TS as a covariate. However, handling cases where source and target datasets have other different covariates is an interesting future work.

**Network architectures.** Meta-GLAR employs an RNN network to compute the representation. This network is composed of two LSTM layers with hidden dimension between 20 and 50 (see Table 8) followed by a linear layer, which allows us to control the dimension of the linear adapation layer independently of the hidden dimension of the RNN. The last LSTM layer has also a residual connection, i.e. the final output is the sum of the output and the input of the layer. Zoneout (Krueger et al., 2016), a form of dropout which works well for RNNs, is applied to both layers with rate 0.1. The same backbone is used for the representation of DeepAR[5] and the RNN auto-regressive model in the ablation analysis, although without the final linear layer which would be redundant in this

---

[3]Oreshkin et al. (2020) use an additional dataset called FRED as the source dataset. However, this dataset is not publicly available and can only be retrieved by crawling an ever changing website over the course of days. Since this procedure does not result in a reproducible dataset, we do not consider it here.

[4]This is a common practice, but becomes crucial when dealing with datasets coming from different domains.

[5]We used the default implementation of GluonTS.

case. For DeepAR and the auto-regressive RNN model, the hidden dimension of both LSTMs is the same and equal to the dimension of the representation. The feedforward network (FF in Figure 1) is composed by two dense layers both followed by ReLU with the same hidden size equal to the dimension of the output. For NBEATS (Figure 2) we used the implementation in GluonTS with sMAPE as loss function, 30 residual blocks with shared weights, and context length equal to 36.

**Training procedure.** Models are trained using a custom version of the GluonTS trainer which uses the ADAM (Kingma and Ba, 2014) optimizer with weight decay and gradient clipping parameters set to $10^{-8}$ and 10 respectively. At the end of training, the final model is the averge of the last 5 checkpointed models (models are checkpointed every 50 ADAM steps). We do not use any early stopping strategy. The loss function used by our method is the sMAPE divided by 100. Learning rate, minibatch size, and number of steps are selected via random search. Minibatches for training are constructed by randomly selecting slices from the time series of the source dataset (Salinas et al., 2020). The slices are split into a context window that is used for fitting the linear adapation model and a prediction window for which the fitted local model is used to generate predictions. The loss for backpropagation is computed on the prediction window. The context window plus prediction window may be longer than some of the shorter time series in the training or test dataset. In this case, the time series is left-padded with 0s to fill the context window and the corresponding time points are not included when fitting the local adaptation layer. This preprocessing is standard in GluonTS and is applied to all methods that we train. To make sure the model can handle such short time series at prediction time, we also include such cases from the training dataset. A hyperparameter controls the minimum number of observations in the context window.

**Hardware.** All models are trained and evaluated on a cloud ml.c5.4xlarge instance with 16 CPU cores and 32GB of RAM. Training and evaluating Meta-GLAR takes around 3-6 hours.

## 6   Conclusion and Future Work

In this work we proposed a novel meta-learning approach for time series forecasting models called Meta-GLAR. Meta-GLAR is trained on a source dataset and can then generate accurate forecasts for new time series that may come from a domain different from the one of the source dataset. This is achieved through a combination of a global deep representation and a local closed-form adaptation layer. Crucially, during training the local closed-form adaption is differentiable, such that the deep representations are learned across multiple time series in a meta-learning fashion, by backpropagating gradients through the solution of the closed-form solver at training time.

We evaluated Meta-GLAR on out-of-sample TS through an extensive empirical study. Results were competitive with NBEATS, the current state-of-the-art method while requiring fewer parameters. Our model was also competitive with classical local methods (e.g. beating the winner of the M3 competition) and outperforms a global RNN-based method with a similar architecture.

While Meta-GLAR did not achieve state-of-the-art results on all datasets, it introduces a novel approach to neural forecasting that draws from meta-learning and global-local forecasting. We showed that including a differentiable local adaptation layer in neural forecasting models can improve forecast accuracy and transfer capabilities. This approach could be used in combination with other neural forecasting backbones such as transformers (Lim et al., 2019; Li et al., 2019).

In contrast to recent results in computer vision (Tian et al., 2020), we showed that differentiable adaptation during training can perform well on forecasting problems. We believe that this is due to the different nature of forecasting datasets, which usually contain a large number of TS/tasks, while few-shot image classification benchmarks are constructed from a large single-task dataset.

We also believe that the present work is an important step towards neural models that combine the ease-of-use of classical forecasting methods (directly used on single time series), with the accuracy gains of deep learning models. However, the scope of applicability of meta-learning forecasting methods remains unclear. For example, further work is needed to remove the assumption that target and source dataset have the same frequency (e.g., daily, monthly, weekly), or to deal with source and target dataset having different dataset or domain specific covariates.

## 7 REPRODUCIBILITY STATEMENT

To ensure reproducibility of our experiments we evaluated our method on publicly available TS datasets and we discussed in detail the training procedure, model parameters, preprocessing, and hardware used in Section 5.1. Furthermore, we state in the Appendix the time lags (Table 7) used by the model, the hyperparameter ranges for the the model selection procedure used in the majority of the experiments (Table 8) and the source-target pairs (Table 6). We also provide the source code for the experiments in the supplemetary material.

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

# Appendix

## A  ADDITIONAL RESULTS

### A.1  TRANSFER FROM M4 (RESULTS BY FREQUENCY)

Table 3-5 contain the results for the models in Table 1 divided by frequency for M3, TOURISM and M4, together with additional baselines. The rows of the tables are divided in 3 sections. The first one contains local models, the second one models with global parameters and the third one models with global parameters which we trained and evaluated.

Meta-GLAR outperforms DeepAR (both trained on M4 using the same random search and evaluation procedure) overall on ELECTR, TRAFF, TOURISM, and M3 and also on most subdatasets of TOURISM and M3. Our method achieves the state of the art on ELECTR, reaching the same ND of ARIMA and NBEATS trained on electricity. Meta-GLAR-ens10 outperforms all the local methods on M3 and is second only to NBEATS. On Tourism, our method performs better than Theta and ARIMA although it falls behind the local method LeeCBaker, which is tailored to tourism TS . On TRAFF Meta-GLAR has a higher ND than NBEATS and local methods but improves substantially over DeepAR trained on M4-HOURLY.

### A.2  ABLATION ANALYSIS ON M4-HOURLY

In Figure 3 we report the results of the ablation analysis on the hourly frequency. Note that in this case, the source dataset, M4-HOURLY, has only $414$ TS. We perform combinations of the following ablations. Replacing the closed-form adaptation with a global linear layer (no Meta in the name). Replacing the closed-form adaptation with a global linear layer only at training time, which is equivalent to fine-tuning the last layer (ADA in the name). Using the observations in the horizon to compute predictions during training instead of iterated forecasts (no ITF in the name).

We note that our method outperforms the autoregressive RNN both for ELECTR and TRAFF. Computing the adaptation only during prediction can be advantageous on TRAFF but significantly degrades the performance on M4-HOURLY. Using iterated forecast during training is beneficial in most cases. We also report the statistics for all the random search runs of the ablations studies in Figure 4. Our methods performs the best in most cases even with random hyperparameters.

## B  ADDITIONAL EXPERIMENTS DETAILS

### B.1  ADDITIONAL BENCHMARK METHODS

We Include the following additional benchmarks methods for an improved comparison. LeeCBaker (Baker and Howard, 2011), a competitive method for tourism TS . EXP (Spiliotis et al., 2019), the state of the art local method on M3. The M4 competition winner (M4 winner) (Smyl, 2020) and the second best entry (Best ML/TS) (Montero-Manso et al., 2020).

### B.2  SOURCE TARGET PAIRS

In Table 6 we describe the source-target combinations that we use for all our experiments. The same combinations are considered by Oreshkin et al. (2020). We use the model trained on M4-QUARTERLY to compute forecasts on M3-OTHER because they have forecast horizons of the same length. ELECTR and TRAFF contain hourly data, hence they are matched with M4-HOURLY.

### B.3  METRICS

To measure the performance of the methods, we use the following metrics.

$$\text{sMAPE}_{\mathcal{D}} = \frac{1}{|\mathcal{D}|} \sum_{(z,x)\in\mathcal{D}} \frac{200}{H_{\mathcal{D}}} \sum_{i=1}^{H_{\mathcal{D}}} \frac{|z_{t_0+i} - \hat{z}_{t_0+i}|}{|z_{t_0+i}| + |\hat{z}_{t_0+i}|} \tag{8}$$

$$\text{ND}_{\mathcal{D}} = \frac{\sum_{(z,x)\in\mathcal{D}} \sum_{i=1}^{H_{\mathcal{D}}} |z_{t_0+i} - \hat{z}_{t_0+i}|}{\sum_{(z,x)\in\mathcal{D}} \sum_{i=1}^{H_{\mathcal{D}}} |z_{t_0+i}|} \tag{9}$$

$$\text{MAPE}_{\mathcal{D}} = \frac{1}{|\mathcal{D}|} \sum_{(z,x)\in\mathcal{D}} \frac{100}{H_{\mathcal{D}}} \sum_{i=1}^{H_{\mathcal{D}}} \frac{|z_{t_0+i} - \hat{z}_{t_0+i}|}{|z_{t_0+i}|} \tag{10}$$

where $\mathcal{D}$ is the TS dataset (for M3, M4, TOURISM we consider each frequency separately), $H_{\mathcal{D}}$ is the corresponding forecast horizon and $|\mathcal{D}|$ is the number of TS in the dataset.

We compute aggregate metrics as follows.

$$\text{sMAPE}_{\text{M3}} = \left( \sum_{\mathcal{D}\in\text{M3}} H_{\mathcal{D}} \times |\mathcal{D}| \right)^{-1} \times \sum_{\mathcal{D}\in\text{M3}} H_{\mathcal{D}} \times \text{sMAPE}_{\mathcal{D}} \tag{11}$$

$$\text{MAPE}_{\text{TOURISM}} = \left( \sum_{\mathcal{D}\in\text{TOURISM}} H_{\mathcal{D}} \times |\mathcal{D}| \right)^{-1} \times \sum_{\mathcal{D}\in\text{TOURISM}} H_{\mathcal{D}} \times \text{MAPE}_{\mathcal{D}} \tag{12}$$

where M3 = {M3-YEARLY, ..., M3-OTHER} and TOURISM = {TOUR-YEAR, ..., TOUR-MONTH}.

Table 3: sMAPE on M3. All models except local ones are trained on M4 except the ones with *target* in the name, which are instead trained on the target dataset.

|  | M3 | M3-YEARLY | M3-QUART | M3-MONTHLY | M3-OTHER |
|---|---|---|---|---|---|
| Theta[*] | 13.015 | 16.900 | 8.960 | 13.850 | 4.410 |
| ARIMA[*] | 14.005 | 17.730 | 10.260 | 14.810 | 5.060 |
| EXP[*] | 12.713 | 16.390 | 8.980 | 13.430 | 5.460 |
| DeepAR-target[*] | 12.671 | 13.330 | 9.070 | 13.720 | 7.110 |
| NBEATS-target[*] | 12.374 | 15.930 | 8.840 | 13.110 | 4.240 |
| DeepAR[*] | 14.767 | 14.760 | 9.280 | 16.150 | 13.090 |
| NBEATS[*] | 12.441 | 15.250 | 9.070 | 13.250 | 4.340 |
| DeepAR-top10 | 13.297 | $16.214 \pm 0.180$ | $9.429 \pm 0.065$ | $14.235 \pm 0.142$ | $4.674 \pm 0.079$ |
| DeepAR-ens10 | 12.884 | 15.640 | 9.228 | 13.782 | 4.536 |
| Meta-GLAR-top10 | 12.746 | $15.842 \pm 0.267$ | $9.282 \pm 0.132$ | $13.513 \pm 0.077$ | $5.033 \pm 0.390$ |
| Meta-GLAR-ens10 | 12.509 | 15.305 | 8.970 | 13.335 | 4.848 |

Table 4: MAPE on TOURISM. All models except local ones are trained on M4 except the ones with *target* in the name, which are instead trained on the target dataset.

|  | TOURISM | TOUR-YEAR | TOUR-QUART | TOUR-MONTH |
|---|---|---|---|---|
| Theta[*] | 20.878 | 23.450 | 16.150 | 22.110 |
| ARIMA[*] | 20.959 | 28.030 | 16.230 | 21.130 |
| LeeCBaker[*] | 19.350 | 22.730 | 15.140 | 20.190 |
| DeepAR-target[*] | 19.276 | 21.140 | 15.820 | 20.180 |
| NBEATS-target[*] | 18.523 | 21.440 | 14.780 | 19.290 |
| DeepAR[*] | 24.787 | 21.510 | 22.010 | 26.640 |
| NBEATS[*] | 18.828 | 23.570 | 14.660 | 19.330 |
| DeepAR-top10 | $23.920 \pm 0.631$ | $26.233 \pm 0.585$ | $17.724 \pm 0.362$ | $25.784 \pm 0.967$ |
| DeepAR-ens10 | 22.889 | 25.174 | 17.000 | 24.640 |
| Meta-GLAR-top10 | $21.254 \pm 0.247$ | $25.806 \pm 0.906$ | $18.072 \pm 0.658$ | $21.418 \pm 0.390$ |
| Meta-GLAR-ens10 | 20.095 | 24.469 | 16.802 | 20.343 |

Table 5: sMAPE on M4 in the non-transfer setting. All models are trained on the target dataset.

|  | M4-YEARLY | M4-QUARTERLY | M4-MONTHLY | M4-HOURLY |
|---|---|---|---|---|
| Best ML/TS[*] | 13.528 | 9.733 | 12.639 | − |
| M4 winner[*] | 13.176 | 9.679 | 12.126 | − |
| DeepAR[*] | 12.362 | 10.822 | 13.705 | − |
| NBEATS[*] | 12.913 | 9.213 | 12.024 | − |
| DeepAR-top10 | $14.100 \pm 0.109$ | $9.942 \pm 0.040$ | $13.160 \pm 0.150$ | $8.982 \pm 0.174$ |
| DeepAR-ens10 | 13.658 | 9.765 | 12.770 | 8.515 |
| Meta-GLAR-top10 | $14.026 \pm 0.119$ | $9.993 \pm 0.036$ | $12.670 \pm 0.061$ | $8.721 \pm 0.110$ |
| Meta-GLAR-ens10 | 13.524 | 9.758 | 12.523 | 7.702 |

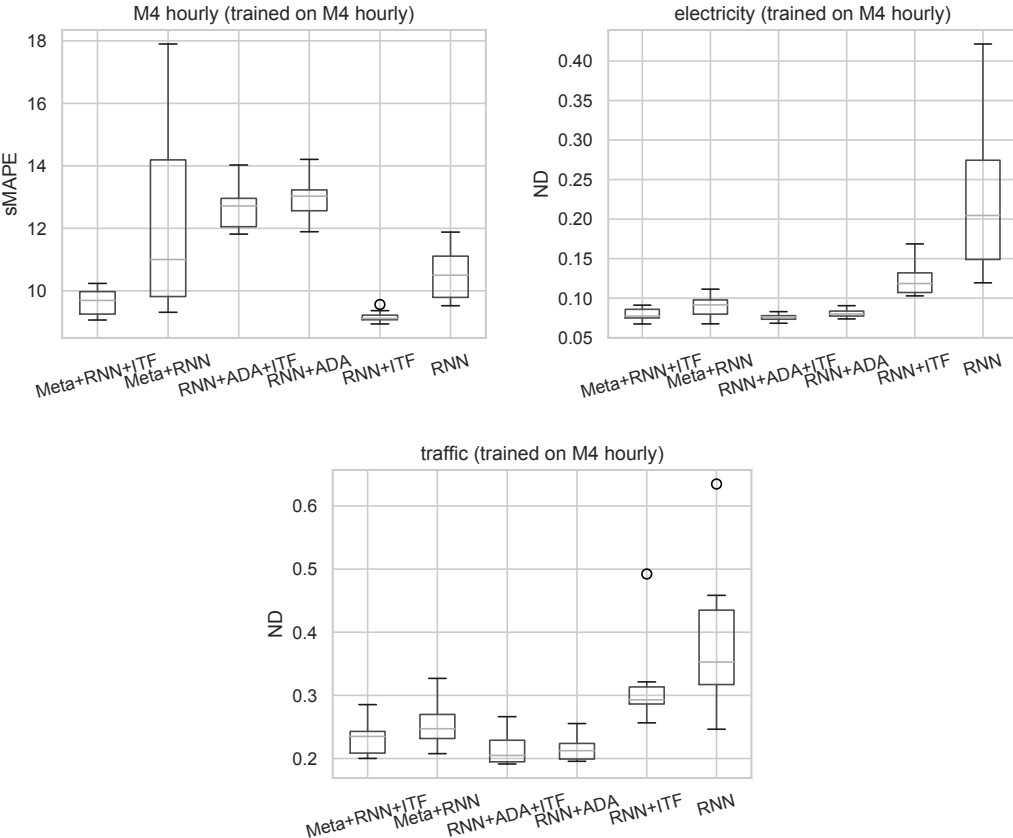

Figure 3: Ablation study for Meta-GLAR (Meta+RNN+ITF) on M4-HOURLY. Ablated elements are the closed-form adaptation from meta-learning (Meta), which is also used only during prediction (ADA) and iterated forecasts during training (ITF). Each box plot show statistics for the 10 over 100 random search combinations with lowest sMAPE computed on 8K random training TS.

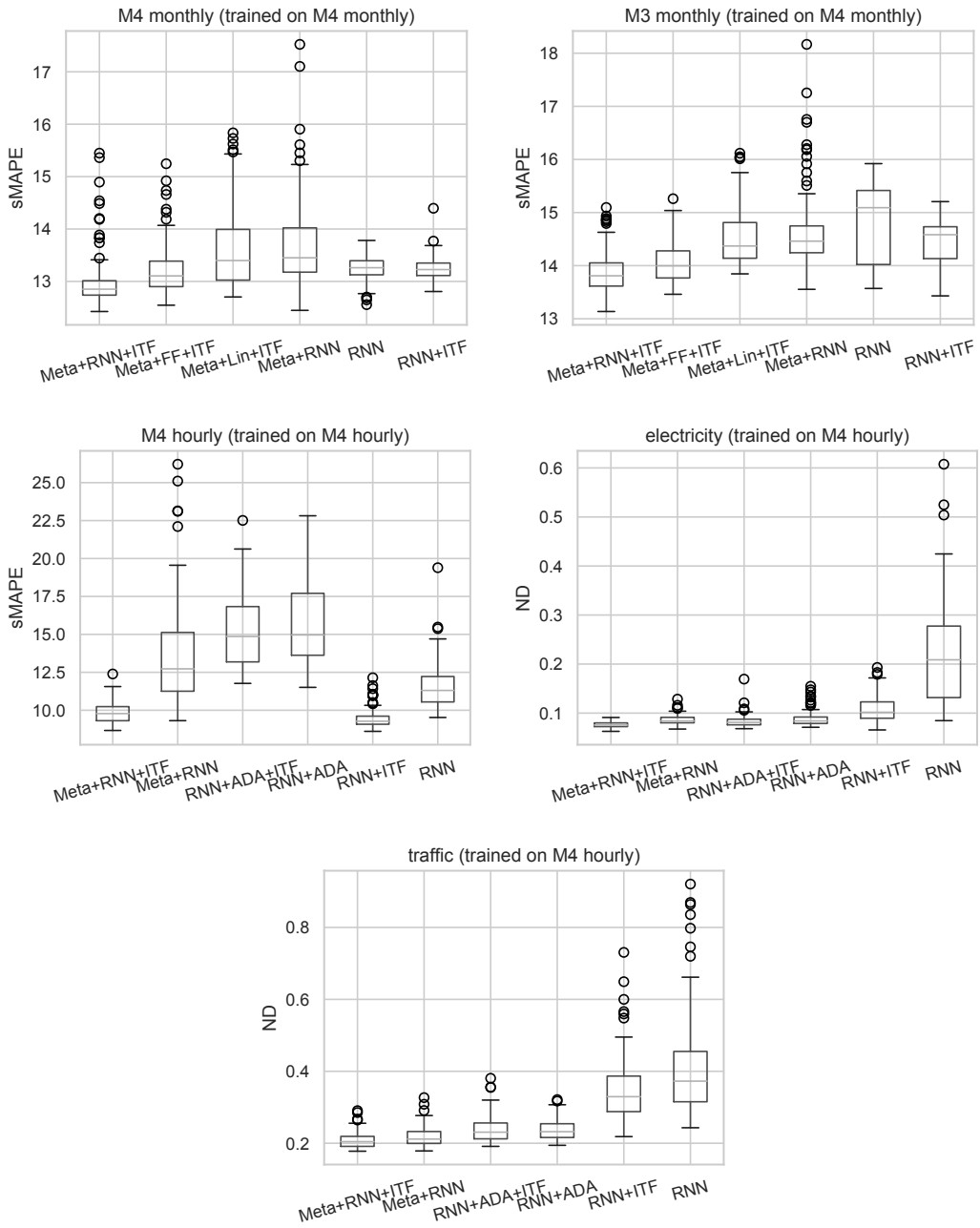

Figure 4: Ablation study for Meta-GLAR (Meta+RNN+ITF). The box plots show statistics for all the runs of the random search (no model selection).

Table 6: Source-targets combination for DeepAR and Meta-GLAR. A model per row is trained on the dataset in the source column and tested on all the ones in the target column. The length of the forecast horizon follows each dataset preceded by a slash. Note that the length of the forecast horizon can differ between source and target.

| Source | Targets |
|---|---|
| M4-YEARLY/6 | M3-YEARLY/6, TOUR-YEAR/4 |
| M4-QUARTERLY/8 | M3-QUART/8, M3-OTHER/8, TOUR-QUART/8 |
| M4-MONTHLY/18 | M3-MONTHLY/18, TOUR-MONTH/24 |
| M4-HOURLY/48 | ELECTR/24, TRAFF/24 |

Table 7: Time lags used by Meta-GLAR, DeepAR and ablations for each time-frequency. These are the defaults in the GluonTS implementation of DeepAR

| Frequency | Lags |
|---|---|
| Yearly | $[1, 2, 3, 4, 5, 6, 7]$ |
| Quarterly | $[1, 2, 3, 4, 5, 6, 7, 8, 9, 11, 12, 13]$ |
| Monthly | $[1, 2, 3, 4, 5, 6, 7, 11, 12, 13, 23, 24, 25, 35, 36, 37]$ |
| Weekly | $[1, 2, 3, 4, 5, 6, 7, 8, 12, 51, 52, 53, 103, 104, 105, 155, 156, 157]$ |
| Daily | $[1, 2, 3, 4, 5, 6, 7, 8, 13, 14, 15, 20, 21, 22, 27, 28, 29, 30, 31, 56, 84, 363, 364, 365, 727, 728, 729, 1091, 1092, 1093]$ |
| Hourly | $[1, 2, 3, 4, 5, 6, 7, 23, 24, 25, 47, 48, 49, 71, 72, 73, 95, 96, 97, 119, 120, 121, 143, 144, 145, 167, 168, 169, 335, 336, 337, 503, 504, 505, 671, 672, 673, 719, 720, 721]$ |

Table 8: Hyperparameters for the random search. The first three entries are respectively the number of steps, minibatch size and learning rate used by the optimizer. *Context mult* multiplies the horizon length of the training dataset to give the final context length for the model. *Representation dim* is the dimension of the input to the last linear layer of the model (which is the dimension of the representation). *Min history length* is the minimum number of observations that must be present in the context window of the training TS slices.

| Name | Type | Values/Ranges |
|---|---|---|
| Number of steps | Categorical | $\{25K, 50K\}$ |
| Minibatch size | Categorical | $\{32, 64, 128\}$ |
| Learning rate | Float | $[1^{-5}, 2^{-3}]$ |
| Context mult | Float | $[0.3, 5]$ |
| Representation dim | Integer | $[20, 50]$ |
| Min history length | Integer | $[24, 100]$ |

