# OpenReview forum: "Meta-Forecasting by combining Global Deep Representations with Local Adaptation"
_ICLR.cc/2022/Conference — ICLR 2022 Submitted_

### Official Review · Reviewer_zvPZ · 2021-11-01

**Correctness:** 2
**Technical Novelty And Significance:** 2
**Empirical Novelty And Significance:** 1
**Recommendation:** 3
**Confidence:** 4

**Main Review:**

This work develops a new method for out-of-sample time-series forecasting (that is, transfer setting). Instead of calling the method a meta-forecasting method, which seems problematic. It would be better to use the more standard and frequently used term of zero-shot transfer learning, which fits much better with the method proposed.

Meta-GLAR is shown to have similar runtime and memory overhead compared to a global one-step ahead RNN with similar structure. However, it is unclear how it compares with the other baselines shown in Table 1? In fact, the only result showing training time I noticed, is in Figure 2 (left). However, this only compares Meta-GLAR to RNN. What about the other methods, especially NBEATS? I was hoping to see a table similar to Table 1, but with training time for all the baseline methods across all the datasets. That said, I would remove the last contribution mentioned, since it is only in terms of RNN, but not the actual state-of-the-art methods.

Deep Factors and the more recent Graph Deep Factors were both shown to outperform DeepAR, NBEATS, and the other baselines used. How do these methods compare to Meta-GLAR? It seems they would be trivial to use in this setting as well. Nevertheless, they should also be included as baselines and discussed appropriately since both utilize global and local components and can be trained with a large corpus of time-series.

The caption of Table 3 and Table 4 have a typo, “Table 3: sMAPE on M3. All All models”. The reproducibility details mentioned in the Appendix are good and easy to follow.


**Summary Of The Paper:**

This work proposes a new forecasting method for jointly learning from a large pool of related time-series. The method called Meta Global-Local Auto-Regression (Meta-GLAR) learns the mapping from representations produced by an RNN to one-step ahead forecasts where the parameters are learned across multiple time-series through backpropagation. This work studies the zero-shot transfer learning problem and proposes a method for it. The method is somewhat incremental and evaluation has some issues, see below for details.


**Summary Of The Review:**

The method proposed is interesting (but somewhat incremental) and the problem is important. However, there are several issues that need to be addressed. The method typically doesn’t perform well compared to the state-of-the-art as shown in Table 1. It is also strange why different metrics are used for different datasets, for instance, of the four datasets used, there are 3 different metrics used in the evaluation ND for Electr/Traff, SMAPE for M3, and MAPE for Tourism. It would have been better to report all 3 metrics for each dataset (or just show one of the metrics, and the others in the appendix). It currently seems to be cherry picked a bit. Furthermore, the training time for each baseline and dataset is missing, please include a table like Table 1 but with the runtime for each baseline and dataset. Please see above for other points. Overall, the contribution and novelty of this work is limited.

---

> ### Author Response · Authors · 2021-11-18
> **Response to reviewer zvPZ**
>
> We thank the reviewer for the useful feedback. We will address their concerns in order.
>
> 1. **Use transfer learning terminology instead of meta-learning.** We admit that it was difficult to find an appropriate name for the method. We decided to use the name Meta-GLAR since the method is effectively a meta-learning method (inspired by similar methods in computer vision) and we think that TS forecasting fits quite well in the meta-learning framework since each TS can be seen as a learning task and TS datasets readily contain multiple time-series (or tasks). Each TS can also be seen as a training example, however this view is limited since a forecasting problem can also be fully specified with a single TS, while the same cannot be done with a single example in supervised learning.
>
>
> 2. **Training time and last contribution.** We added and discussed the training time for NBEATS in the learning curves paragraph of the experiments section. NBEATS trains faster than our method since it enjoys a greater level of parallelism. However, we also show in the same paragraph that prediction times are quite similar (for the monthly frequency). We argue that in our setting, training time is less important, since the model would be trained less often and would ideally not require retraining for new time-series. Concerning the last contribution, we added an explanation of the results of the ablation study before talking about the computational costs. We argue that since our method can in principle be seen as an augmented one-step ahead RNN (only the last layer differs), comparing the time complexity with the standard RNN is more informative, since for both models the computational cost of computing the representation is the same and the additional cost comes entirely from the closed adaptation layer. Furthermore the computational cost of the standard RNN we use is very similar to DeepAR, since both share the same backbone. We added a comment on this in the learning curves pargaraph of the manuscript.
>
>
> 3. **Deep Factor and Graph Deep Factor baselines.** The mentioned models have been shown to outperform DeepAR and NBEATS when trained and evaluated on the same set of TS. While this may be interesting to study, there is no evidence that these methods will perform well in our transfer setting where source and target TS datasets differ. Furthermore since those are global-local methods they also learn TS-specific parameters for each TS in the training set which will be useless when forecasting an unseen TS. Hence, they will not be trivial to use in our setting since they would have to re-learn the TS-specific parameters. Please see also the second point of the general response.
>
>
> 4. **Typo in Caption of Table 3 and 4.** Thanks for pointing this out, we corrected it in the revision.
>
>
> 5. **The method does not perform well compared to the state of the art.** It is true that we only outperform NBEATS in 1 over 4 datasets. However we argue that the gap is quite close for M3 and our method always outperforms DeepAR, which shares the same backbone and is also a one-step ahead model. Given the simplicity and generality of our approach and the lower number of parameters compared to NBEATS we argue that this is a remarkable result worth it to be published at this venue.
>
>
> 6. **Cherry picked metrics.** We did not cherry pick the metrics. Please see the first point of the general response for more details.
>
>
> 7. **Training time table is missing.**  We do not claim to have faster training time w.r.t. other baselines and we believe including a comprehensive time table for all methods is outside of the scope of the work. We already provide the training and prediction time comparison with NBEATS (added in the revision) and with the RNN sharing the same backbone. We also commented that the RNN time is very similar to the one of DeepAR, since they have a similar architecture. See also point 2. Our method can be seen as an augmented RNN but we show that the increased complexity of the last layer has only a marginal impact on training and prediction times which are dominated by the computation of the representation from the one-step ahead RNN (see learning curves paragraph in the experiments section).

---

### Official Review · Reviewer_FZbR · 2021-11-01

**Correctness:** 2
**Technical Novelty And Significance:** 2
**Empirical Novelty And Significance:** 2
**Recommendation:** 3
**Confidence:** 3

**Main Review:**

Strengths
---
1.	The automatic output layer tuning they propose is an interesting idea. Although not explored in this paper, this could potentially be adopted to cope with concept drifts or structural breaks which are prevalent in non-stationary time series datasets.

Weaknesses
---
However, I do have some concerns with the paper in its current form, which has a couple of areas that need to be addressed.
1.	The authors use a couple of technical terms in non-standard ways, making the introduction slightly confusing:
  * Out-of-sample -- in the time series forecasting domain, out-of-sample data typically refers to data forwards in time that is unavailable during training. Generalisation forwards in time is the primary purpose of training forecasting models, whether statistical or machine learning approaches. The authors use this to refer to generalisation performance on a different time series entirely, however – which depends critically on similarities between source and target tasks. For instance, it is not unreasonable for a forecasting model that predicts patient survival times to fail to predict retail sales.
  * Multi-task learning – global forecasting models are typically not referred to as multi-task learning, as the goal is to achieve good forecasting performance across all entities and time steps, as measured by a single evaluation metric. See [1] for more on multi-task learning.
2.	Suitability of benchmarks – while local (statistical) models can be useful for **data-limited** settings, I am not sure these are suitable comparisons for zero-shot models which cannot access the target dataset before prediction time. Assuming some data is available for training, there are few global network options available for small data regimes (e.g. [2]), and comparisons to other standard transfer learning/domain adaptation techniques should be evaluated. Similarly, any differences vs NBEATS could have arisen from the choice of encoder -- a NBEATS style encoder should be used within the proposed meta-learning framework for a fair comparison.
3.	How does the framework handle transfer learning between datasets with a different number of covariates? The utility of the approach would very limited if only univariate target inputs can be used, as a lot of useful information would also need to be discarded.
4.	Complexity of matrix inversion -- assuming that h(t) has dimensions d, inverting the dxd matrix typically is O(d^3). As it is possible for d > t (see optimal hidden state sizes in DeepAR and Temporal Fusion Transformer vs encoder length), this can be greater than the sequential matrix multiplications for the LSTM (O(d^2 t) I believe). As such, computational complexity may indeed be a concern in some scenarios for this layer.

References
1.	Crawshaw. Multi-task learning with deep neural networks. arXiv:2009.09796.
2.	Rangapuram et al. Deep State Space Models for Time Series Forecasting. NeurIPS 2018.


**Summary Of The Paper:**

The authors propose a new meta-learning framework to tackle the zero-shot learning problem for time series data – through the combination of:
1.	A standard autoregressive architecture encoding historical information (e.g. DeepAR).
2.	An adaptive linear output layer whose weights are calibrated in closed-form using encoder history.

This allows the model to be trained end-to-end on a source task, while automatically performing domain adaptation when applied to a new target task.


**Summary Of The Review:**

While the raw idea has promise, many corrections/clarifications need to be made before it is ready for publication -- specifically related to the terminology used, whether inputs between datasets can differ, and the complexity claims of their proposed layer. In addition, more suitable benchmarks are required to fully evaluate performance claims.

---

> ### Author Response · Authors · 2021-11-18
> **Response to reviewer FZbR**
>
> We thank the reviewer for the useful feedback. We will address their concerns in order.
>
> 1. **Terminology.** We agree that some of the terminology is ambiguous or may mean different things to different communities. We modified the second paragraph of the intro by clarifying what we mean with out-of sample TS, by removing the term multi-task when referring to global models and by adding the following phrase at the top. *From a machine learning perspective, each TS represents a forecasting task.*
>
>
> 2. **Suitability of benchmarks.** We argue that classical methods are very effective in the meta-learning setting and that the global method we picked for the comparison (DeepAR) is a natural choice since it shares the same RNN backbone as our method. Standard Transfer learning and domain adaptation techniques usually require access to more TS in the target dataset and even when using a single one they are often computationally expensive (for example fine-tuning the whole network requires multiple gradient steps at prediction time). See also the second point in the general response. **On Using our method with the NBEATS backbone for a fair comparison**, our approach and NBEATS are substantially different and our method with the NBEATS backbone would be significantly different from NBEATS. In our opinion, this hybrid method will not provide a fair comparison with NBEATS and should be treated as a separate method. Combining the two will also be redundant, since the NBEATS backbone is already designed to adapt to the context window as our closed-form last layer.
>
>
> 3. **Dataset with different covariates.** We do not handle dataset or domain specific covariates and we agree that this can be an interesting future direction. We added the following text at the end of the input/output section of the paper. *In our setting it is counterproductive to use the index of the TS as a covariate. However, handling cases where source and target datasets have other different covariates is an interesting future work.* We also added a comment on this in the last paragraph of the conclusions.
>
>
> 4. **$O (d^3)$ complexity of matrix inversion.** We agree that this might be a concern. However we use $d \leq 50$ in our experiments and we show that the closed form adaptation does not have a great impact in the total training and prediction times (see learning curves paragraph in the experiments section). Furthermore if $t_0 < d$ one could solve the ridge regression problem inverting a $t_0 \times t_0$ matrix instead to reduce the computational cost further. In either case, solving the linear system directly instead of inverting the matrix has a lower cost and could speed up computation even further (which is the lower bound we report in the text).

---

### Official Review · Reviewer_Tvo1 · 2021-11-10

**Correctness:** 2
**Technical Novelty And Significance:** 2
**Empirical Novelty And Significance:** 1
**Recommendation:** 5
**Confidence:** 3

**Main Review:**

## Strengths

* The authors describe the problem setting for meta forecasting as well as the Meta-GLAR approach quite clearly.

* The experiments conducted on Meta-GLAR are quite comprehensive (large hyper-parameter sweeps, comparisons with models directly trained on target TS). Furthemore, the experimental setup is described in sufficient detail along with supplementary code to ensure reproducibility.

* The ablation study conducted to determine the importance of each component in the Meta-GLAR model is comprehensive and shows that each component (RNN, Meta closed-form adaptation, and iterated forecasts) in the model adds to the performance although it does not clarify why RNN with iterated forecasts leads to worse performance than an RNN baseline.

## Weaknesses

* The novelty of the approach itself is quite limited. The local model (*differentiable local adaptation layer*) is essentially a linear layer applied on top of the RNN in the context window. The Meta-GLAR approach could be (approximately) described as training an RNN on the full time series without training the linear layer in the forecast horizon, and then using the linear weights from the context window only during transfer to new tasks.

* In Table 1, why did the authors chose to report three different metrics across the four datasets (ND on ELECTR and TRAFF, sMAPE on M3, MAPE on M4) in their comparisons? If all three metrics were calculated across all baselines and the authors' proposed models then these results need to be included (at least in the appendix). Since the authors have not reported their reasoning behind chosing different metrics for different datasets, it raises the suspicion that these comparisons might be cherry picked.

* In their comparison with fine-tuning, did the authors try fine-tuning both the final layer and RNN? That seems like a fairer comparison versus keeping the RNN model fixed and only training the final layer. The authors used closed form solutions for the final layer which might get more expensive to compute if the RNN model wasn't fixed. A comparison with a global linear layer fine-tuned alongside the RNN model would provide a better picture of the importance of their local adaptive layer.

* One broader question that this research raises is about the scope and relevance of the meta learning framework to TS forecasting. The framework lends itself very well to domains like computer vision due to relationship between visual tasks [1]. However, it is not immediately obvious why it would be relevant to time series forecasting where the underlying domains and granularities can vary widely. I'd like to see the authors comment on the scope and limits of this work in their claims:
> Meta-GLAR is trained on a source dataset and can then generate accurate forecasts for new time series that may **differ significantly** from the source dataset


## Errata/Clarifications

*  Did the authors mean to say generalization in the Into (para 2)?
> Global-local approaches ..., exhibit a greater level of **specialization** as they learn parameters that are shared by all TS in the training set.
* The reference regarding the use of closed form solvers in spatial regression seems incorrect in the Related Work (end of para 2)
> ...and spatial regression (Iwata and Kumagai, 2020), while we are not aware of any application in forecasting.

## References

[1] Taskonomy: Disentangling Task Transfer Learning. Amir R. Zamir, Alexander Sax, William Shen, Leonidas J. Guibas, Jitendra Malik, Silvio Savarese; Proceedings of the IEEE Conference on Computer Vision and Pattern Recognition (CVPR), 2018,

**Summary Of The Paper:**

The authors propose an autoregressive framework, Meta-GLAR, comprising of a global RNN model and a local linear model (learned using a closed form solution) for the task of meta-forecasting of time series (TS). The linear model is the task-specific model varies for each TS in the dataset, whereas the RNN model is the meta model that is shared across al time series. The authors perform comparisons with local models as well as NBEATS and DeepAR, and show that Meta-GLAR is competitive with some of them on particular metrics. They also perform an ablation study to show that each component in their model is important to the overall zero-shot transfer in TS forecasting.

**Summary Of The Review:**

The problem of zero-shot transfer in time series forecasting is important but it is also necessary to define its scope. The method presented by the authors is novel but incremental. There are concerns regarding the reporting of results (see Weaknesses) which cast doubt on the model performance, which is already lower than NBEATS on some tasks. Overall the paper presents an interesting approach to performing meta-forecasting but it needs more detailed analysis in terms of its applicability as well as completeness in terms of reporting its results. I would be willing to consider an improvement in the score if these issues are addressed.

**Post-rebuttal:** My concerns regarding scope and cherry-picking have been sufficiently addressed in the revised submission and the authors' comments. However, I am still concerned about the novelty of the work as well as the extent of ablations. I have increased my overall score to 5 (marginally below the acceptance threshold) and provided additional feedback to the authors in comments.

---

> ### Author Response · Authors · 2021-11-18
> **Response to reviewer Tvo1**
>
> We thank the reviewer for the useful feedback. We will address their concerns in order.
>
> 1. **Limited novelty.** We disagree with the reviewer. The novelty of our work resides mainly in how we learn the RNN parameters. In particular, we disagree with the reviewer's statement *The Meta-GLAR approach could be (approximately) described as training an RNN on the full time series without training the linear layer in the forecast horizon, and then using the linear weights from the context window only during transfer to new tasks.*. This approach would only learn features for fitting the model well in the context window and possibly perform poorly on the forecast horizon due to overfitting.
> At each training step, we compute an explicit local adaptation by solving an optimization problem in closed form in each TS context during the forward pass. However, **we also compute the loss over the TS horizon and backpropagate through the closed-form solution** to compute the gradients w.r.t. the representation and regularization parameters. A similar approach has been originally proposed for meta-learning in computer vision, but it has not been previously studied  in time series forecasting. We believe that the simplicity of the approach compared with other meta-learning methods like MAML is one of its major strengths and definitely not a weakness, since it makes the method easy to understand and also to implement.
>
>
> 2. **Cherry-picked metrics** We did not cherry pick the metrics, please see the first point of the general response for more details.
>
>
> 3. **Fine-tuning the whole network** We did not try to fine-tune the whole network. We agree that this could be interesting, although more costly and probably harder to tune than only adapting the last layer, but outside of the scope of this work. See also the second point of the general response.
>
>
> 4. **Scope and relevance of the meta-learning framework for TS forecasting** We thank the reviewer for the interesting comment. We believe the meta-learning framework is relevant to tackle time-series forecasting, since at least in a single domain or dataset there can be hundreds or thousands of different time-series (or tasks) sharing important structure like seasonality, domain and even exogenous variables. Moreover, in most domains it is reasonable to think that there will be new, unseen time series to forecast, e.g. the electricity demand of a new user or the price of the stock of a new company. However, we agree that for TS coming from different domains the advantages of meta-learning are less clear. In Oreshkin et al (2020) and partially in our work it is shown that, contrary to global neural forecasting models, meta-learning models trained on M4 can perform very well on electricity, traffic and tourism, despite the datasets being from different domains. Nonetheless, the gap between local models and meta-learning approaches is not that wide even when source and target TS datasets are in the same domain, and it is still not clear whether a single meta-learning model can deal with different seasonalities or different covariates.   We updated the last paragraph of the conclusion and future work section commenting on this point. We also modified the reported claim in the conclusion by changing *TS which may differ significantly* to *TS which may come from a domain different from the one of the source dataset.*
>
>
> 5. **Did the authors mean to say generalization in the Intro (para 2)?** No, we meant specialization: global-local approaches have parameters which are specific to each of the training TS, thus we can say they specialize to forecast those.
>
>
> 6. **Wrong reference**. Thanks for pointing this out, we corrected the reference in the updated manuscript.

---

> > ### Comment · Reviewer_Tvo1 · 2021-11-29
> > **Response to authors**
> >
> > Thank you for taking the time to address each of the concerns individually, as well as explaining your perspective on the novelty and scope of your work. After reading through your comments and the revised submission, I have upgraded my overall rating to 5 (marginally below the acceptance threshold). My concerns regarding scope and cherry-picking have been sufficiently addressed in the revised submission and the authors' comments. However, I am still concerned about the novelty of the work as well as the extent of ablations. I feel the novelty is still incremental, the ablations performed regarding individual components of Meta-GLAR are not very conclusive, and further results from a transfer learning approach are needed for a fairer comparison regarding the utility of a meta learning approach.
> >
> > I feel the authors could have either (a) proposed a novel approach to meta forecasting and demonstrated its applications to generalization, or (b) adapted a known meta-learning approach from other fields (e.g. vision or NLP) and performed extensive experiments and controlled ablations to understand the usefulness of such an approach to time series forecasting. The paper in its revised form falls somewhere between these two possible studies and as a result is not a strong enough contribution for the audience at ICLR. I hope the reviewers are able to use this feedback to improve upon their work in a revised submission.

---

### Author Response · Authors · 2021-11-18
**Response to all reviewers**

We thank all reviewers for their feedback. Given the very low ratings we understand that we have failed to get our point across and made several small changes (in blue) in the revised manuscript to make the paper clearer. We still disagree with the evaluation. We explain why first by addressing the reviewers’ most common concerns in this general response and then by responding to each reviewer individually.

1. **Cherry-picked Metrics.** Both reviewers Tvo1 and zvPZ mention that we might have cherry picked metrics for Table 1 since we use 3 different metrics without explanation. **We did not cherry pick the metrics but chose them to have a fair comparison with previously published results which mainly use the metrics we picked**. Since this was not clear, we added the following paragraph in the experimental section of the revised manuscript. *Following Oreshkin et al (2020), to compare with previous results from the literature we report 3 different metrics for the datasets: sMAPE for M4 and M3; ND, which corresponds to the P50 loss in Salinas et al (2020), for ELECTR and TRAFF; and MAPE for TOURISM.* Additionally, we also did not cherry-pick the benchmark datasets but used all of the ones by Oreshkin et al (2020) except FRED, which was hard to retrieve and is using a prohibitive license. We believe that the benchmark we provide is quite comprehensive, containing 5 datasets, 4 frequencies and several different domains.


2. **Comparison with other baselines.** All the reviewers mention other possible baselines. In particular,  they mention fine-tuning the whole network (Tvo1) and other neural models like  DeepState (FZbR) and (Graph) Deep Factor (zvPZ). **We compare against the SOTA methods for the meta learning benchmark. While we agree that trying out DeepState and DeepFactor in the meta learning setting is interesting future work, this is outside of the scope of this work**. These neural models are usually trained and tested on the same TS dataset which is often homogeneous and contains hundreds of TS, and often rely on dataset specific covariates like the index of the TS. There is little indication they will perform well in our setting where it is more difficult to use dataset specific covariates and we train models on a heterogeneous dataset and test them on each TS of a different dataset individually.  In addition, Deep Factors and Grap Deep Factors are global-local models which train global and local parameters jointly. Adapting them to our setting is not trivial as it will require to re-learn the local parameters on each time-series. **Fine-tuning the whole network on each TS can be interesting but will face similar issues of other meta-learning methods like MAML and Reptile**, namely the requirement to set additional hyperparameters (step size and number of gradient descent steps) and the increased computational cost of running gradient descent at test time for each TS. We added a comment on this in the baselines paragraph of the experimental details section. **We believe that we benchmarked our method against very competitive baselines**. In particular, NBEATS, which is the state of the art for all datasets; the local models ARIMA and Theta which are among the best local methods in our setting, theta was indeed the winner of the M3 competition. We also compare our method with the RNN in the ablation and with DeepAR, using the same HPO procedure as our method since they have very similar hyperparameters due to them having the same RNN backbone.

---

### Decision · Program_Chairs · 2022-01-20

**Decision:**

Reject

**Comment:**

This paper proposes an autoregressive framework that combines RNN and local linear component for the problem of meta-forecasting of time series. The linear model can domain-adapt to different time series while the RNN component is shared across series. Reviewers thought the problem was important, the paper was generally clear and the experiments extensive. However they found the significance to be limited and all took issue with some of the ways that the comparisons were done. FZbR also raised the issue of complexity of the matrix inversion component of the method.  I believe this paper does fall on the rejection side of the fence due to the issues of complexity, significance and evaluations. With some development, the paper could certainly be ready for acceptance.